# Increased COVID-19 Testing Rates Following Combined Door-to-Door and Mobile Testing Facility Campaigns in Oslo, Norway, a Difference-in-Difference Analysis

**DOI:** 10.3390/ijerph182111078

**Published:** 2021-10-21

**Authors:** Kristin Hestmann Vinjerui, Ingeborg Hess Elgersma, Atle Fretheim

**Affiliations:** 1Division for Health Services, Norwegian Institute of Public Health, 0213 Oslo, Norway; IngeborgHess.Elgersma@fhi.no; 2Centre for Epidemic Interventions Research, Norwegian Institute of Public Health, 0213 Oslo, Norway; Atle.Fretheim@fhi.no; 3Faculty of Health Sciences, Oslo Metropolitan University, 0130 Oslo, Norway

**Keywords:** COVID-19, SARS-CoV-2, humans, impact evaluation, non-pharmaceutical interventions, difference-in-difference

## Abstract

High testing rates limit COVID-19 transmission. Attempting to increase testing rates, Stovner District in Oslo, Norway, combined door-to-door campaigns with easy access testing facilities. We studied the intervention’s impact on COVID-19 testing rates. The Stovner District administration executed three door-to-door campaigns promoting COVID-19 testing accompanied by drop-in mobile COVID-19 testing facilities in different areas at 2-week intervals. We calculated testing rates pre- and post-campaigns using data from the Norwegian emergency preparedness register for COVID-19 (Beredt C19). We applied a difference-in-difference approach using ordinary least square regression models and robust standard errors to estimate changes in COVID-19 testing rates. Door-to-door visits reached around one of three households. Intervention and comparison areas had identical testing rates before the intervention, and we observed an increase in intervention areas after the campaigns. We estimate a 43% increase in testing rates over the first three days following the door-to-door campaigns (*p* = 0.28), corresponding to an additional 79 (95% confidence interval, −54 to 175) people tested. Considering the shape of the time series curves and the large effect estimate, we find it highly likely that the campaigns had a substantial positive impact on COVID-19 testing rates, despite a *p*-value above the conventional levels for statistical significance. The results and the feasibility of the intervention suggest that it may be worth implementing in similar settings.

## 1. Introduction

In Norway, the test-isolate-trace-and-quarantine strategy has been a key part of the public health response to the COVID-19 pandemic. High testing rates are essential to break chains of infection. During the first wave of the pandemic, the national strategy only encouraged testing of symptomatic individuals with a high risk of exposure, due to low test capacity. In subsequent waves, everyone with any symptom of COVID-19 has been encouraged to take the test [1].

As elsewhere, the pandemic in Norway has affected socioeconomic groups and geographical areas unequally, most evident in the capital, Oslo. Until February 2021, foreign-born in Norway both had higher rates of confirmed COVID-19 infection and lower testing rates, compared to the general population [2]. Barriers to testing can be related to language skills and health literacy, as well as technological and financial shortcomings, and these barriers are greater among immigrants [3,4,5].

The city district of Stovner has the highest proportion of immigrant inhabitants in Oslo and has experienced high rates of COVID-19 cases. In response, the local authorities have implemented several measures. In November 2020, they appointed local youth as “corona hosts”, providing hand sanitizer, face masks and multilingual information at central locations. In January 2021, they introduced free of-charge drop-in testing services at the city district centre, and in February 2021, they carried out three rounds of door-to-door visits where they encouraged everyone to be tested. The campaigns were accompanied by the nearby placement of mobile COVID-19 testing facilities.

While the foreign-born population is a recognized vulnerable group, the local efforts to strengthen infection control has had a broad approach. This is in line with Norwegian traditions as a social democratic welfare state [6], characterized by a rights-based, universal benefit scheme. It is shown that generous policies associate positively with health for the total population, including vulnerable subgroups [7].

There are few studies of the impact of non-pharmaceutical interventions to limit COVID-19 disease spread [8,9], and we are not aware of any studies of measures to increase COVID-19 testing rates. We saw an opportunity for evaluating the combination of running a door-to-door campaign and providing easy access to COVID-19 testing facilities by collaborating closely with the Stovner District administration in their rollout of the intervention, and utilizing data from the Norwegian emergency preparedness register for COVID-19.

The aim of this study was to describe COVID-19 testing rates pre- and post-intervention and to assess to what extent the universal door-to-door campaigns promoting COVID-19 testing accompanied by mobile testing facilities affected COVID-19 testing rates.

## 2. Materials and Methods

### 2.1. Setting

Stovner is one of 15 city districts in the capital of Norway, Oslo. It is further divided into 23 basic geographical units, which are constructed by Statistics Norway. These area units are expected to be homogeneous with respect to the natural environment, economic conditions, communication, and building structure. In January 2021, 33,279 people were registered as living in Stovner [10], of which 39% were immigrants and 21% had immigrant parents [11]. Most homes are in apartment blocks (63%) [12]. From late February 2020 to the end of January 2021, 4.9% of the Stovner District population had been registered with a positive COVID-19 test [13] (p. 9, Figure 5), the highest proportion of all city districts in Oslo. During the time of the study, the proportions of COVID-19 cases per 100,000, calculated in two-week intervals, generally increased in the Stovner District, as shown in Figure 1 [14] (Dates: 9 February, 16 February, 23 February, 2 March, 9 March, and 16 March 2021).

### 2.2. Contents of the Intervention (Treatment)

Having the highest COVID-19 incidence rate in Oslo prompted the Stovner District administration to increase local efforts for infection control. The COVID-19 testing intervention was shaped by local ideas, input from state authorities, and common sense. It consisted of door-to-door campaigning accompanied by mobile testing facilities. While the migrant population was an established vulnerable group in Stovner, the intervention was, for feasibility reasons, universal in its form.

Figure 2 depicts how the intervention was executed in three rounds, separated by 2-week intervals, in different basic geographical units. The intervention was introduced with a one-day door-to-door campaign with canvassers recruited from the district administration. As door-to-door visits hold the potential to increase the risk of COVID-19 transmission, the canvassers strictly followed standard precautious measures to prevent spread. They provided oral and written information encouraging everyone to get a COVID-19 test, and about the free-of-charge, drop-in mobile testing facility which was placed in the same geographical area for one or more days following the door-to-door campaign. In the figure, the dots represent the order of the rounds and the placement of the mobile COVID-19 testing facilities, while the shaded areas show in which basic geographical units the canvassing took place. The presence of the mobile testing facilities in the three rounds varied for economic and logistic reasons. The canvassers left written information in the mailboxes of the households that were not reached.

In detail, on 1 February 12 canvassers covered three basic geographical units from 10:30 am to 6:00 pm. The mobile COVID-19 testing facility was present on days 1 and 3 after the door-to-door visits. On 15 February 12 canvassers covered one basic geographical unit from 10:00 am to 4:00 pm and the testing facility was present on days 1, 3, and 5. The third door-to-door campaign involved nine canvassers on 1 March from 10:00 am to 3:30 pm and covered two basic geographical units in the district centre. Here, a mobile COVID-19 testing facility had already been present daily for several weeks prior to the door-to-door campaign.

The selection of areas for the intervention to take place was pragmatic, mainly based on the presence of apartment blocks, since this facilitates reaching more households over a shorter period. The district administration and researchers collaborated in the planning of the rollout to ensure that the campaigns took place within basic geographical units. This was important since basic geographical units were the lowest level of individual geographical detail available in the data we had access to.

### 2.3. Data Source

In response to the pandemic, the Norwegian emergency preparedness register for COVID-19 (Beredt C19), was established by the Norwegian Institute of Public Health in April 2020 [15]. The register includes both real-time and historical information on residents of Norway, from numerous data sources. The national identity number enables individual linkage of data but is anonymous to the analysts. For privacy reasons, the registry does not include addresses, only the basic geographical unit for each resident. In this study, we used data from the Norwegian Surveillance System for Communicable Diseases and the National Population Register.

### 2.4. Variables

The outcome variable was COVID-19 testing rates. We calculated this from daily updates on individually performed COVID-19 PCR tests from the Norwegian Surveillance System for Communicable Diseases. We required permanent residency to establish a stable denominator to calculate testing rates. The National Population Register provided information on permanent residence in basic geographical units, as well as age, sex, households, and data on country of birth. We defined foreign-born as a person born outside Norway and Norwegian born as someone born in Norway.

### 2.5. Study Population and Follow-Up

Our study population includes every person permanently residing in a basic geographical unit in the Stovner District. We included 32,717 people, excluding 61 individuals missing information on the basic geographical unit. The sample had complete information on age, sex, and country of birth. The data on households is from 2020 and was missing for 929 persons. Between 25 January and 8 March, we followed the study population for 15 days in three rounds, 7 days prior to, on the day of, and 7 days after the door-to-door campaigns encouraging COVID-19 testing.

### 2.6. Statistical Analysis

For both descriptive and statistical analysis, in all rounds, the comparison group included all basic geographical units in Stovner District that had not previously been exposed to the intervention.

To evaluate time trends in testing rates, we calculated the proportions tested per day in the comparison and the intervention areas, 7 days before to 7 days after, separately for each round and also combined the data from all three rounds by defining the days of the door-to-door campaigns promoting COVID-19 testing as a common day zero. Finally, we stratified the combined data in Norwegian- and foreign-born to evaluate if the time trends in testing rates varied in sub-populations.

To assess the effect of the complete intervention on COVID-19 testing rates, we combined individual-level data on testing, sociodemographic covariates, and on the basic geographical unit of residence from 7 days before to 7 days after the door-to-door campaign took place, which gave us a panel dataset, with person-day as the unit of analysis. We performed difference-in-difference analyses, comparing changes in testing rates from 8 days before (including the day of the door-to-door campaign) to 7 days after the intervention, between the total 6 basic geographical units where the interventions took place, and the remaining 17 basic geographical units serving as the comparison group.

We generated ordinary least square regression models for two time periods, days 1 to 3 and days 4 to 7. As the number of basic geographical units is low, we used robust standard errors, calculated with Wild bootstrap [16,17], clustered on the basic geographical unit. In addition to a crude model, we adjusted for age, birth country, and round of intervention. The following equation shows the model we estimated. $Treat_{i}$ is a dummy of whether the individual i is in an intervention area, $Post_{1-3}$ is a dummy for days 1 to 3 after the intervention, and $Post_{4-7}$ is a dummy for days 4 to 7 after the intervention. Intervention (treatment) status and time is interacted, making up the difference-in-difference estimators, $\delta_{1}$ and $\delta_{2}$. In the protocol we declared that $\delta_{1}$ was the primary coefficient of interest in the analysis. X is a vector of control variables.
$$Y_{it}=\beta_{1}Treat_{i}+\beta_{2}Post_{1-3}+\beta_{3}Post_{4-7}+\delta_{1}\left(Treat_{i}\times Post_{1-3}\right)+\delta_{2}\left(Treat_{i}\times Post_{4-7}\right)+\beta_{4}X+\varepsilon_{it}$$

The relative increase in testing was calculated by computing the predictive marginal effects of the difference-in-difference estimator and computing the relative change between intervention and the non-intervention subjects. The absolute increase in testing was calculated by multiplying the coefficient of the difference-in-difference estimator with the number of people in the intervention areas and the number of days the treatment effect was modelled to persist (three days).

We used R to perform the statistical analysis.

#### Additional Descriptive Analysis

Prior to the introduction of the door-to-door campaigns encouraging COVID-19 testing, the Stovner District administration placed a mobile testing facility in one basic geographical unit for three separate days during one week in late January. We repeated calculations of testing rates per day in the week before and after the first day of introducing the mobile testing facility with the remaining 22 basic geographical units as a comparison group to evaluate time trends in the testing rates provided the mobile testing facility only.

## 3. Results

### 3.1. Descriptive Results

In total, 32,717 individuals in 14,400 households were included, of which nearly one third resided in the intervention areas (Table 1). The demographic characteristics of the intervention and comparison populations were similar, except for a higher proportion of foreign-born individuals in the intervention group. The intervention group contributed 141,285 person-days and the comparison group 1,189,305 person-days.

### 3.2. Implementation and Time Trends in Testing Rates

There was no formal registration of contact rates, i.e., the proportion of households that were reached through the door-to-door campaign, but the canvassers estimated around one in three. The number of COVID-19 tests conducted in the designated mobile facilities varied. In rounds 1 and 2, 76 and 97 tests were performed in the week after the door-to-door campaign promoting COVID-19 testing. In round 3, the COVID-19 testing facility was present also in the weeks prior to the door-to-door campaign. Here, 494 tests were performed in the 8 days before (including the day of the door-to-door campaign) and 727 tests in the 7 days after canvassing. In Figure 3A–C we present the testing rates per day for the intervention and comparison areas in the week before and after the door-to-door campaigns for each round, in chronological order.

For each round, the level and trend of COVID-19 testing rates were quite similar in all intervention and comparison areas prior to the door-to-door campaign. After canvassing, the testing rates were greater in the intervention areas on days 1 and 3 in the first round (Figure 3A) and on day 1 in the second round (Figure 3B). In the third round, the testing rates increased in both the intervention and the comparison areas, and on day 3 the testing rate was higher in the non-intervention areas (Figure 3C).

We combined the data from all three rounds, by defining the days of each door-to-door COVID-19 testing campaign as day zero (Figure 4).

Combined, the COVID-19 testing rates were close to indistinguishable between the intervention and comparison areas, prior to the door-to-door COVID-19 testing campaigns. On day 1, there was an increase in testing rates in the intervention areas, which was not seen in the comparison areas. The following days, the testing rates were similar in the two groups.

In Figure 5 we present the same data as in Figure 4, separated into Norwegian- and foreign-born inhabitants. There was practically no difference in levels or time trends in testing rates.

We found little or no difference in testing rates when we compared the one geographical unit (Stig) where a pop-up mobile COVID-19 testing facility was placed for three days in January, without an accompanying door-to-door campaign encouraging COVID-19 testing, with the other areas in the Stovner District (Figure 6).

### 3.3. Main Results

All three rounds of door-to-door campaigning accompanied by mobile testing facilities combined yielded a difference-in-difference in testing rates between the areas with and without the intervention, before and after canvassing, of 0.28% (95% confidence interval (CI) −0.19% to 0.62%) for the first three days following the door-to-door campaigns (Table 2). Adjusting for age, birth country and round of intervention did not alter the estimate. The relative increase was 43%, which translates to an additional 79 (95% CI, −54 to 175) people tested. The estimate for days 4 to 7 showed no change in the effect of the intervention on testing rates between the intervention and comparison areas.

## 4. Discussion

On average across the three sites, door-to-door campaigns to encourage COVID-19 testing were associated with a non-significant absolute difference of 0.28% at nearby free drop-in mobile testing facilities over the first three days after canvassing. We estimated a 43% relative increase in testing rates in the intervention areas. Findings were similar for Norwegian- and foreign-born participants. We acknowledge that the number of basic geographical units yield a small sample size and hampers our ability to draw conclusions with a high degree of statistical certainty. Still, the shape and consistency of the time series curve serves as convincing evidence of a causal relationship between the intervention and change in testing rates.

The effects of non-pharmaceutical interventions have been greatly under-explored during the COVID-19 pandemic [8,9]. To our knowledge, few, if any studies have assessed the impact of interventions to increase COVID-19 testing rates. Difference-in-difference analysis is a well-recognized method to study causal relationships in the public health setting [18,19], but infectious disease dynamics pose challenges to this approach [19,20]. In our study, the key assumption, i.e., that pre-intervention trends of the outcome were parallel in the intervention and comparison groups [18,19], seems to be met. Further, the COVID-19 testing rates appear stable and at similar levels, which increases the reliability of the difference-in-difference model [19]. The intervention was performed in one city district, in which the population characteristics are relatively homogeneous and we can reasonably assume a similar prevalence of COVID-19 across various parts of the district, implying that the intervention and non-intervention groups are broadly comparable also in the pandemic setting [19]. While the sample size limits our ability to draw conclusions based on conventional statistical significance levels, the use of an acknowledged analytical method where basic requirements appear fulfilled, the striking shape of the time series curves and the size of the effect estimate imply that the intervention did increase testing rates. The public health relevance of having 79 additional people tested will depend on several factors, including the incidence of COVID-19 and how much emphasis decision-makers put on controlling transmission. Fortnightly local transmission rate ranged from 196 to 1090 per 100,000 during the study period.

The intervention was not based on any explicit theory of behavioural change, and we did not investigate mechanisms for changes in testing behaviour. Overall, the campaign relied on local knowledge and community engagement, which are seen as key elements in the successful development and implementation of public health interventions [21]. Combining qualities of belonging and expertise, using in-group health personnel, was used to target the Somali population in Oslo during the first wave of the pandemic [5]. Posting information videos in private social media channels seemed promising to increase the understanding of and compliance to precautious measures [5]. Universal door-to-door campaigns are used in response to poliovirus outbreaks [22] and were applied during the Ebola epidemic in Liberia in 2014–2015 [23]. During the Ebola outbreak, canvassing was proposed to increase adherence to precautionary measures reinforced by the use of local intermediaries as they were perceived as monitorable and accountable [23]. Door-to-door visits may influence behaviour through establishing social connections, enabling the canvassers to adapt the information to the target group. There may also be a social pressure aspect. Improving accessibility is an established population-level strategy to influence behaviour [24], through for instance increased availability and affordability [25]. The largest increase in testing rates was found after the first and second round, where the door-to-door campaign and mobile COVID-19 testing facilities were both new and introduced simultaneously. The increase in testing rates was smaller in the final round, where a mobile testing facility had been available for 5 weeks prior to the door-to-door campaign, possibly indicating a ceiling effect. Furthermore, we found little or no change in testing rates in the area where a mobile testing facility was placed without a door-to-door campaign promoting COVID-19 testing. We caution against putting too much weight on these observations, but one interpretation could be that introducing door-to-door canvassing and easy access to testing simultaneously may have the strongest impact on testing behaviour.

A major strength of our study is that the implementation in time and space, as well as highly valid data from Beredt C19, made it possible to obtain detailed time curves and thereby assess the impact on changes in COVID-19 testing rates over a full week. The local campaign was feasible and unaffected by the analyses, which were swift, and the findings led the Norwegian Institute of Public Health to recommend the intervention be continued and expanded in the city of Oslo [26].

The main methodological limitations of our study are the small number of geographical units and a non-experimental design, which leaves some degree of uncertainty about whether the associations we observed represent causal relationships. Additionally, we cannot discard the possibility of spillover effects, i.e., that the intervention also affected testing behaviour in the surrounding comparison areas. This may have diluted our findings and yielded an underestimate of the intervention’s effectiveness. Finally, we did not assess the results of the COVID-19 tests. Any positive tests could have partly explained local increases in testing rates during the study period.

Our findings show that introducing door-to-door campaigns promoting COVID-19 testing and easy access to testing facilities is a promising strategy that may be worthwhile to consider in comparable settings where it is feasible to implement. Potential improvements of the intervention should be explored, like when in the day to conduct door-to-door visits to improve contact rate and whether daily accessibility to the mobile testing facilities would yield a higher impact on testing rates.

## 5. Conclusions

COVID-19 testing rates differed 0.28% in difference before and after, in the areas with and without, door-to-door campaigns encouraging testing accompanied by mobile testing facilities. In the intervention areas, testing rates increased 43%. Findings were non-significant (*p* = 0.28), limited by the small sample size of 23 areas, while the time series curves depict a clear relationship between the intervention and change in testing rates. The intervention proved feasible and ought to be transferable to similar urban areas. The literature is scarce on evaluations of non-pharmaceutical interventions to limit the spread of COVID-19. This study demonstrates that close collaboration between researchers and local administrators may make evaluating the impact of such interventions possible, and that it can be done quickly with results that are useful to decision-makers. Increased knowledge on interventions to increase testing rates may also be relevant in future pandemics.

## Figures and Tables

**Figure 1 ijerph-18-11078-f001:**
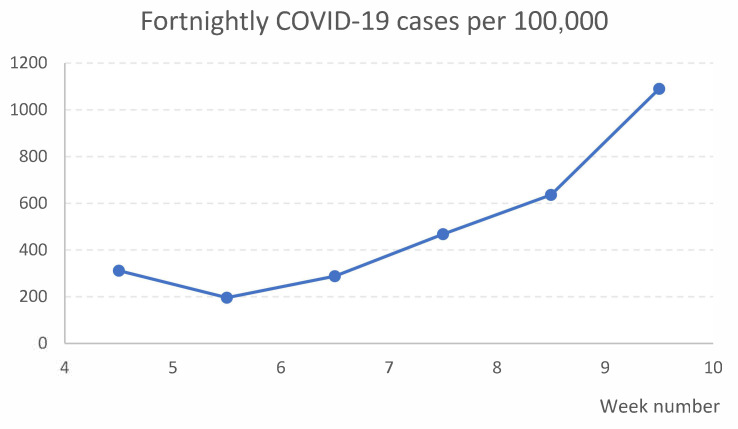
Biweekly calculated number of COVID-19 cases per 100,000 in the Stovner District, from 25 January to 14 March 2021.

**Figure 2 ijerph-18-11078-f002:**
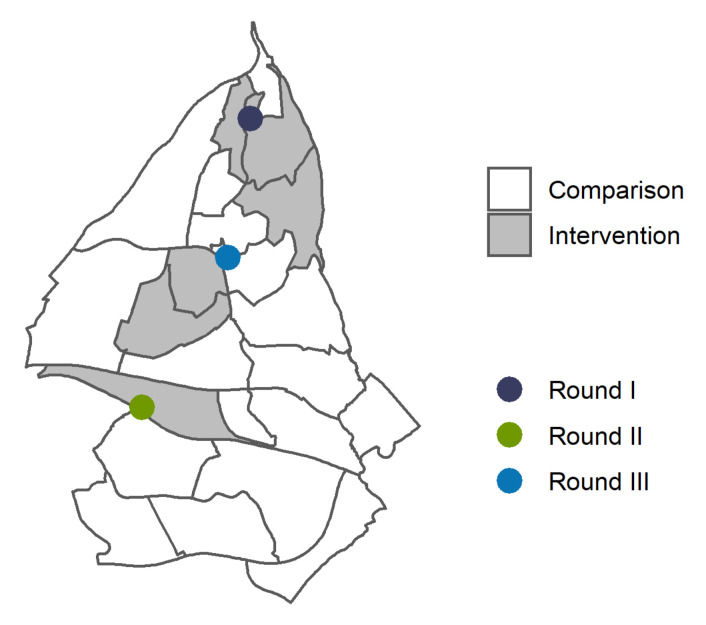
Map of Stovner District. In grey, intervention areas and in white, comparison areas. The dots mark the location of the mobile testing facility and show the order of the campaigns.

**Figure 3 ijerph-18-11078-f003:**
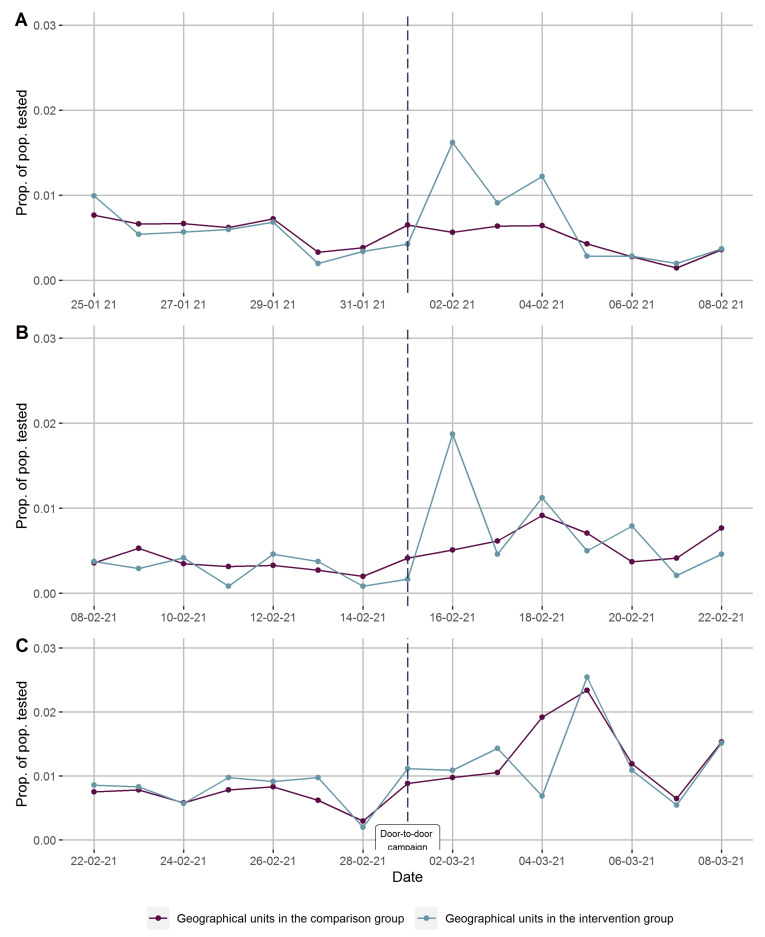
(**A**–**C**) Proportions of the population tested for COVID-19 before and after three separate door-to-door campaigns promoting COVID-19 testing (day 0) accompanied by mobile testing facilities in geographical units stratified on intervention. A: first round, B: second round, and C: third round.

**Figure 4 ijerph-18-11078-f004:**
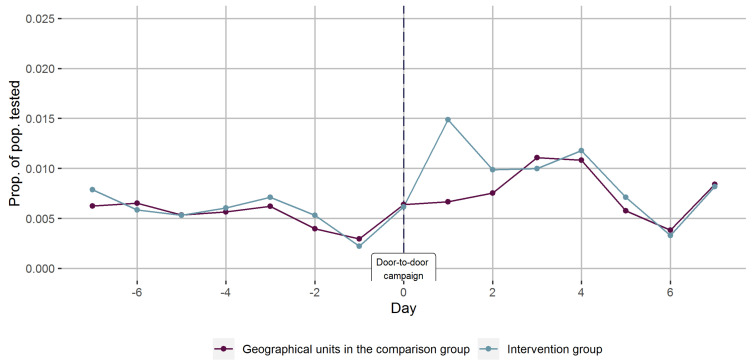
Joint proportions of the population tested for COVID-19 before and after totally three door-to-door campaigns promoting COVID-19 testing (day 0) accompanied by mobile testing facilities in geographical units stratified on intervention.

**Figure 5 ijerph-18-11078-f005:**
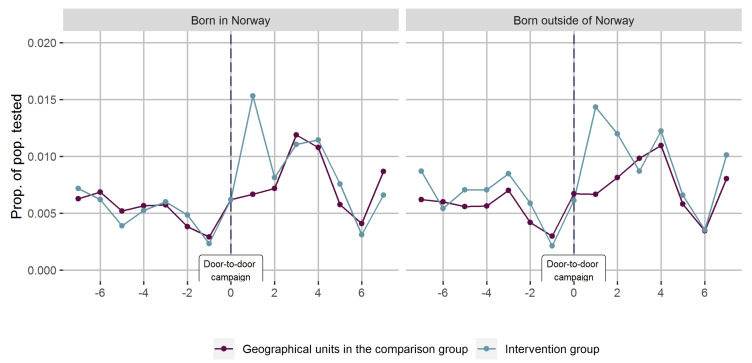
Joint proportions of the population tested for COVID-19 in geographical units with and without door-to-door campaigns promoting COVID-19 testing (day 0) accompanied mobile testing facilities from, in total, three interventions, stratified by Norwegian and foreign-born.

**Figure 6 ijerph-18-11078-f006:**
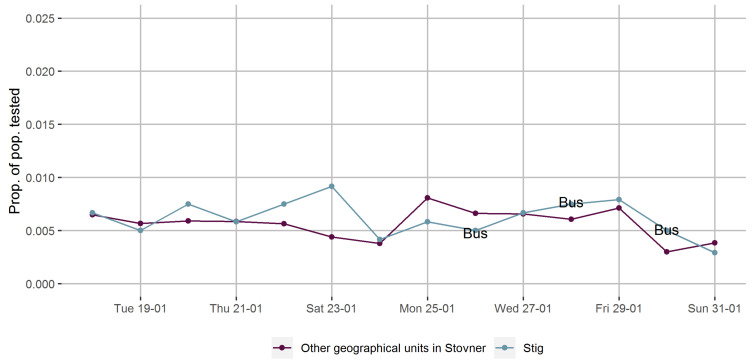
Proportions of the population tested for COVID-19 in geographical units with (purple) and without (blue) a mobile COVID-19 testing facility (bus).

**Table 1 ijerph-18-11078-t001:** Description of the complete sample stratified in the intervention and comparison groups.

Variable	Intervention Group		Comparison Group		Total	
	Mean	SD	Mean	SD	Mean	SD
Age, yr.	37.2	23	38.6	23	37.9	23
	Frequency	Proportion	Frequency	Proportion	Frequency	Proportion
Geographical units	6	26%	17	74%	23	100%
Households	4244	29%	10,156	71%	14,400	100%
Individuals	9419	29%	23,298	71%	32,717	100%
Female	4757	51%	11,486	49%	16,243	50%
Born abroad	4257	45%	8574	37%	13,414	41%
Person-days	141,285	11%	1,189,305	89%	1,330,590	100%

Footnote: Abbreviations: SD, standard deviation; yr., years.

**Table 2 ijerph-18-11078-t002:** Results from regression models estimating the effect of three door-to-door campaigns promoting COVID-19 testing accompanied by mobile testing facilities on COVID-19 testing rates.

	Model 1			Model 2		
DiD-Estimator	Estimate	95% CI	*p*-Value	Estimate	95% CI	*p*-Value
$\delta_{1}$ (days 1–3)	0.0028	(−0.0019, 0.0062)	0.28	0.0028	(−0.0019, 0.0062)	0.28
$\delta_{2}$ (days 4–7)	0.0000	(−0.0061, 0.0047)	0.97	0.0000	(−0.0061, 0.0047)	0.97

Model 1: Crude model, intervention, time period (days 1 to 3 and days 4 to 7 after the door-to-door campaign) and interaction of the intervention and time periods. Model 2: Model 1, and adjusted for age, country of birth and round of the campaign. Abbreviations: DiD, difference-in-difference; CI, confidence interval; $\delta_{1}$ the interaction of the intervention and days 1 to 3; $\delta_{2}$ the interaction of the intervention and days 4 to 7.

## Data Availability

The individual-level data used in this study is not publicly available due to privacy laws.

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
