# Peer review of "Increased COVID-19 Testing Rates Following Combined Door-to-Door and Mobile Testing Facility Campaigns in Oslo, Norway, a Difference-in-Difference Analysis"

_ijerph, 2021, doi:10.3390/ijerph182111078_

Round 1

Reviewer 1 Report

Please expand the introduction by providing more information on the importance of reaching out particular vulnerable populations such as migrants.

The presentation of the results can be improved perhaps with subheadings describing the different rounds of interventions.

In the discussion, please also make reference to similar interventions targeting migrant communities to provide some strategies comparison. This would help to support the results of the present study, which are limited by the small sample size.

Author Response

Thank you for the review and particular considerations for the population born outside Norway.

We agree that migrants are a vulnerable group in this pandemic. In the introduction, we state that foreign-born has had higher rates of confirmed cases and lower testing-rates, than the general population, which implies it’s importance, but we have added some references and arguments as follows, to make this even clearer:

Introduction, line 44: “…, and these barriers are greater among immigrants (Indseth, 2021; Kjøllesdal, 2021; Brekke, 2021)”

We also added a paragraph on how a universal intervention, is a typical Norwegian response, which is expected to have impact on the groups with higher needs as well, see:

Introduction, line 53: " While the foreign-born population is a recognized vulnerable group, the local efforts to strengthen infection control has had a broad approach. This is in line with Norwegian traditions as a social democratic welfare state (Esping-Andersen, 1990), characterized by a rights-based, universal benefit scheme. It is shown that generous policies associate positively with health for the total population, including vulnerable sub-groups (Bergqvist, 2013)."

Turning to presentation of the results, we have added subtitles to separate descriptive and statistical results, lines 235, 244 and 323: 3.1. Descriptive results; 3.1.1. Implementation and time trends in testing-rates; and 3.2. Main results

As for the discussion, it was very challenging to find any literature on universal or targeted initiatives to increase testing-rates. We would much appreciate if you know any relevant literature. As now stated in the Introduction, shown in our sample, and added to the first paragraph of the Discussion, lines 362 to 364 “The door-to-door campaigns to encourage COVID-19 testing accompanied by free-of-charge drop-in mobile testing facilities was followed by a large rise in testing-rates in the intervention areas, similarly among Norwegian- and foreign-born.”, the universal approach also reaches more vulnerable groups. There was a targeted, social media information campaign towards the Somali population in Oslo, spring 2021. We have added to the Discussion, line 394: «Combining affiliation and health expertise, was used to target the Somali population in Oslo during the first wave of the pandemic (Brekke, 2021). Posting information videos in private social media channels, seemed promising on increased understanding of and compliance to pre-cautious measures (Brekke, 2021).” 

Reviewer 2 Report

In their discussion, the authors state : "Finally, we did not assess the results of the COVID-19-tests. Any positive tests could have partly explained local increases in testing-rates during the study period"

Would it be possible to stratify results by positive/negative tests results. This would be on interest and strengthen the researcher's results/conclusions. 

Otherwise this was a fantastic paper and is an important contribution to COVID-19 research.

Author Response

We are very pleased that you find our study of great value in COVID-19 research.

We agree that the positive test-rate is an important measure. Unfortunately, it was not included in the original protocol and due to strictly regulated use of the Norwegian emergency preparedness register for COVID-19, we are not able to meet this request.  Since positive test-rates depend largely on the local transmission status, we are confident that our chosen outcome, testing-rates overall, is a good measure for increased infection control in all contexts.

Reviewer 3 Report

The authors set out to test the effect of a community-based intervention on COVID-19 testing in Norway.

My comments:
(1) The manuscript would benefit from a careful review and edit for detail as interpretation of the findings can be difficult. For example, they mention a person id number and registration without explaining how these influence their data. Are people without ID illegal immigrants? Are unregistered people visitors - who has to be registered. People not from Norway cannot interpret this and the information seems to be important to properly define the population. Similarly, the cohort of 32954 (presumably out of 33279) indicates that most residents were included - but I am not sure if this means that their households were contacted and how successfully. Since there was no record of success it is difficult to interpret the conversion rate and the success of the study. But assuming 1/3 success, as quoted, we have 11,000 people contacted, presumably many are living in the same household. Of those, there is an increase of 82 (around 0.7%) although the CIs are -42 to 170 so this is not significant. Again, more detail of how households relate to individual numbers etc would be beneficial. Also, the quoted 4.9% covid test positives - is that incidence, point prevalence, period prevalence?
(2) On a related note, assuming that there was a significant increase in testing, given the very very transient effect, would this be economically efficient? This could be answered only if we knew the result of the tests. If out of 82, they had found NO cases, then perhaps not. If they had found 10 cases, then perhaps yes. It is difficult to judge this without this information. Ultimately, these programs will only be conducted if their economic and public health usefulness can be demonstrated.
(3) Finally, I am not entirely certain why Norwegian- and non-Norwegian born individuals were separated. This could be useful as a subanalysis to test if different approaches are required for these populations but this is not really argued by the authors.

Given the lack of marked statistical significance, a very transient effect, and (without testing) uncertain public health significance, I am not convinced that this paper delivers a clear message that can help inform public policy.

Author Response

Thank you for a thorough review, with important focus on details and clarifications to ease interpretation of the findings. We have replied aiming to add to background knowledge and at the end of each comment, we summarize the changes made to the manuscript.

1) The personal, national ID number is assigned to every Norwegian at birth and to immigrants who plan to stay in Norway for more than six months and have a residence permit.  Only permanent residents were available in the pandemic registry with information on residing in a city district, which is also necessary to establish a stable denominator to calculate testing rates. People visiting Norway or have a temporary residence permit or are illegal immigrants, were excluded from the statistical analysis.  The number of permanent residents in Stovner by large outnumber visitors, temporary or illegal residents and should not influence the results of this study.

The explanation for the lower number of people included in the study (32717 people), than registered as living in the city district Stovner (33279), can be that the latter is the official number by 1 January, while our sample was from February-March. Note that the number of people is lower than originally reported (237 fewer). The data source, the National pandemic emergency register links several data sources, which are all continuously updated, but at different intervals. The lower number of individuals when repeating the analysis in September vs March, likely reflects a lag in data on people moving from the area. 61 individuals missed information on residence (basic geographical unit) and was excluded. The statistical analyses were little changed and characteristics of the population (table 1), is unaffected by the minimal loss of individuals included.

We understand that we should stratify the sample, to inform the reader of the population size in the intervention groups (everyone living in the total 6 basic geographical units) and the comparison groups (everyone in the remaining 17 area units). For this, we have added a table of demographic characteristics in the intervention and comparison group, including number of households in the Results section, table 1.

Assuming 1/3 success in contacting 9419 individuals, one can assume 3140 may have had the intervention. The point estimate of 79 additional people tested, then implies 2,6% of those contacted.

To your final question in comment 1, the quoted 4.9% COVID-19 test positives, is the period prevalence. We added to the Methods/Study population, line 77: “From late February 2020 to the end of January 2021 4.9% of the Stovner District population were registered with a positive COVID-19 test[9] (p. 9, figure 5), ….”

With this discussion in mind, we rewrote the Methods section, moving Study population towards the end, after Data Source and Variables, with an aim to comprise and clarify, please see the manuscript for the complete changes.

2) We agree it is important to evaluate cost/effectiveness of the intervention and find the positive test-rate an important measure. Unfortunately, it was not included in the original protocol and due to strictly regulated use of the Norwegian emergency preparedness register for COVID-19, we are not able to meet this request. Still, positive test-rates depend largely on context (the local transmission status), and we are confident that our chosen outcome, testing rates overall, is a good measure for increased infection control in all contexts and thus of public health relevance.

3) We stratified the population in Norwegian and non-Norwegian born, in describing time trends to detect if the universally shaped intervention also reached the groups that would benefit most from it. The findings were promising and while sub-analyses would have been interesting, we chose to only perform statistical analysis of the complete sample, because the statistical power was low. We understand that the reasons for stratification could have been better highlighted and have added to the manuscript, both in the Introduction, lines 44 and 53 to 58; Methods/Statistics, line 193; and Discussion 364, to tie this perspective better together.